# Asymmetric Lip Hyperpigmentation in a Transplant Patient

**DOI:** 10.3390/dermatopathology12040041

**Published:** 2025-11-10

**Authors:** Vincent Kimpe, David Alvarez Martinez, Sébastien Menzinger, Gürkan Kaya

**Affiliations:** 1Department of Dermatology and Venereology, University Hospital of Geneva, 1205 Geneva, Switzerland; david.alvarezmartinez@hug.ch (D.A.M.); sebastien.menzinger@hug.ch (S.M.); gkaya@hug.ch (G.K.); 2Clinical Pathology Department, University Hospital of Geneva, 1205 Geneva, Switzerland

**Keywords:** hyaluronic acid filler, drug-induced hyperpigmentation, post-inflammatory hyperpigmentation, dermatopathology

## Abstract

A 56-year-old patient presented to our dermatology clinic with asymmetric hyperpigmentation on her lower lip, which had developed over the previous six to twelve months. Her medical history included kidney and pancreas transplants, requiring chronic immunosuppression, and two lip filler injections with hyaluronic acid (HA). Clinical examination revealed irregular pigmented macules limited strictly to the lower lip. Histological analysis showed epidermal melanosis, pigmentary incontinence, solar elastosis, and amorphous dermal HA deposits, without evidence of melanocytic hyperplasia or granulomatous inflammation.

## 1. Introduction

A 56-year-old woman presented to our dermatology clinic, concerned about darkening on her lower lip, which had developed over the previous six to twelve months. Gradually, irregular dark spots appeared without any bleeding, discomfort, or other symptoms. The patient’s medical history included two organ transplants—a kidney and pancreas transplant in 2015 and a second kidney transplant in August 2024. She had been receiving long-term immunosuppressive therapy, including ciclosporin, everolimus, and several other drugs between January 2022 and November 2023. Among those, the patient received amlodipine, hydrochlorothiazide and valsartan which are documented in the literature as potentially causing medication-related hyperpigmentation.

The patient used to expose herself extensively to ultraviolet (UV) irradiation without protection while sunbathing until her first organ transplant. Since then, she has been more cautious and uses sun protection. She also reported receiving two injections of hyaluronic acid (HA) filler in both lips, one in 2020 and the other in 2022, with no immediate side effects after these procedures. In the corresponding period, she was taking immunosuppressive drugs, including prednisone, everolimus, tacrolimus and ciclosporin. She denied any family history of comparable pigmentation or other syndromes, such as gastrointestinal polyposis.

Clinical examination revealed irregular, asymmetric, hyperpigmented macules on the lower lip, without any noteworthy cellular infiltration. The remainder of the oral mucosa was normal. Most of the face and other uncovered areas of the skin were notable for stigmata of chronic sun exposure, with mottled hyperpigmentation and solar lentigines (see Figure 1A for the clinical presentation). She showed no hyperpigmentation in palmar folds or other cutaneous areas. Nails were similarly unaffected.

The histological analysis of a biopsy showed a flattened and atrophic epidermis with massive dermal solar elastosis, associated with amorphous grayish deposits in the mid and deep dermis (Figure 1B). Fontana-Masson staining revealed a heterogeneous epithelial hyperpigmentation and mild pigmentary incontinence, whereas colloidal iron staining highlighted the intensely stained blue amorphous dermal deposits (Figure 2A,B). Since this staining is used to identify acidic mucopolysaccharides, glycoproteins and other mucosubstances in tissues, and therefore to confirm the presence of HA in tissue samples, such as biopsies from skin injected with HA fillers, these deposits were consistent with HA filler material. Melan-A immunostaining confirmed the absence of a melanocytic hyperplasia or an atypical melanocytic proliferation (Figure 2C).

## 2. What Is the Diagnosis?

(a) Drug-associated hyperpigmentation.

(b) Solar lentigo.

(c) Melanotic macules.

(d) Post-filler hyperpigmentation of the lip.

(e) Laugier-Hunziker syndrome.

## 3. Diagnosis

(d) Post-filler hyperpigmentation of the lip.

## 4. Discussion

When considering drug-induced hyperpigmentation, several distinct mechanisms can be involved, or any combination of them. Pigmentation changes can occur through direct deposition or accumulation of substances in the dermis, through modulation of melanin production in melanocytes, or through photosensitization that leads to secondary hyperpigmentation. According to several recent systematic reviews, four of the drugs taken by our patient could be related to cutaneous changes: amlodipine, hydrochlorothiazide, ciclosporin, and prednisone. Both ciclosporin and prednisone are not specifically linked to patchy hyperpigmentation, and the level of proof seems to be low [1,2]. Amlodipine can produce grayish blue to brownish reticulate hyperpigmentation affecting the face and sun-exposed areas, usually after six to twelve months of treatment. Only two amlodipine-associated hyperpigmentation cases have been documented, although the specific mechanism was unclear. In both, hyperpigmentation was more extensive and systemic than in our patient, covering sun-exposed zones such as the face, neck, and forearms. Mucosal manifestations on the tongue and palate were described as slate-gray or grayish blue discoloration [3,4]. When associated with telmisartan, hydrochlorothiazide may cause grayish or brown photo-distributed hyperpigmentation, affecting the face, back, dorsal feet and forearms [2]. Nevertheless, hyperpigmentation in association with hydrochlorothiazide has also been explained through photosensitization in phototoxic eruptions [5,6]. Similarly to a normal UV response, a phototoxic reaction promotes the release of IL-1α, which stimulates melanotrophin production by keratinocytes [7]. In contrast with these cases of drug-associated hyperpigmentation, our patient presented asymmetrical beige to brown macules, strictly limited to the lower lip, making drugs unlikely culprits for her symptoms.

Classically, solar lentigo is histologically characterized by epidermal hyperplasia with clubbing, basal hyperpigmentation and melanocytic hyperplasia without atypical melanocytic proliferation, usually associated with solar elastosis. The irregularly pigmented macules on our patient’s lip, together with numerous stigmata of chronic sun exposure, make this a plausible diagnosis. Some lesions of solar lentigo may show a flattened epidermis, which remains hyperplastic. However, the histopathological analysis of the lesion of our patient showed an atrophic epidermis, which is unusual for this diagnosis. Moreover, the context and history of our patient, with previous HA filler injections at the sites of the lesions, led us to discuss different mechanisms of post-filler hyperpigmentation.

Post-inflammatory hyperpigmentation (PIH) is a recognized but relatively rare complication of HA injections, occurring more frequently in patients with darker skin types (Fitzpatrick IV–VI) or in cases of significant trauma related to the injection technique. Several mechanisms may be involved. First, the direct mechanical trauma caused by the injection itself, particularly in cases of multiple puncture sites or the use of thick cannulas, can stimulate melanocyte activity and induce excessive melanin production [8,9]. Alternatively, in type IV hypersensitivity or delayed hypersensitivity reactions, T lymphocytes secrete pro-inflammatory cytokines, leading to macrophage recruitment and activation and maintenance of inflammation. This chronic inflammation stimulates melanocytes’ melanin production, leading to hyperpigmentation [8]. Furthermore, the degradation of HA, particularly low molecular weight fragments (LMW-HA), has also been associated with inflammation through the activation of macrophages and dendritic cells, leading to a release of pro-inflammatory cytokines that may promote pigmentation. It is important to note that PIH can still be seen clinically and histologically after the disappearance of inflammation. The presence of contaminants in HA fillers, such as residues of bacterial proteins or endotoxins from the manufacturing process, could also induce immune activation and low-grade chronic inflammation, exacerbating hyperpigmentation [10]. Other possible causes of chronic inflammation are biofilms, structured microbial communities with a protective matrix adhering to non-organic surfaces. Bacteria in these biofilms are often resistant to antibiotic treatments and can evade immune surveillance [11]. Persistence of these germs could sustain low-grade inflammation and thereby stimulate melanocyte activity. It is important to distinguish actual pigmentary changes from the Tyndall effect, a bluish discoloration caused by the diffusion of light through an HA injected too superficially, the latter being reversible with hyaluronidase, whereas PIH can persist for several months [8,9]. Finally, it is possible for telangiectasias to develop through neovascularization on the skin above the injection site of the filler. This can occur as a result of higher skin tension, forceful rubbing, or even the use of steroids to address post-injection erythema [8]. The Tyndall effect and neovascularization were not consistent with the clinical presentation in our patient.

Our patient’s status as a solid-organ transplant recipient, on several long-term immunosuppressive drugs, may have influenced the local inflammatory response to filler injections. Immunosuppressive agents modify inflammatory and reparative pathways and are associated with altered wound healing and an increased risk of cutaneous complications after procedures [12,13]. For instance, immunosuppression could plausibly dampen an otherwise florid inflammatory reaction, with only low-grade or subclinical inflammation sufficient to drive epidermal melanosis and pigment incontinence [14]. Moreover, individual drugs have heterogeneous effects on pigment biology and repair; in vitro experimental data indicate that cyclosporine can decrease melanocyte tyrosinase activity, whereas inhibitors of the mTOR pathway have been linked to impaired cutaneous repair in rats [15,16]. Taking these considerations together, it is plausible that our patient’s immunosuppressive regimen modulated the clinical and histologic expression of a filler-related reaction, although we evidently cannot establish causality.

Filler-induced hyperpigmentation often presents with irregular, asymmetrical, and localized pigmentation [9,17]. Histopathological features may include epidermal melanosis and mild pigmentary incontinence, exactly as seen in our patient’s biopsy. The lighter appearance of the central vermilion most likely reflects the « normal » lip color associated with major actinic damage and solar elastosis as seen in Figure 1B, rather than a true hypopigmentation, as we did not observe a decrease in epidermal melanin with the Masson Fontana stain at the periphery of the biopsy. Even in the absence of overt postprocedural inflammation—which may remain subclinical—low-grade inflammatory processes can stimulate melanocyte activity, ultimately leading to PIH. The clinicopathologic spectrum of dermal filler complications is very broad, ranging from foreign body granulomatous reactions to subtle inflammatory responses with minimal clinical findings [18]. Studies of HA fillers further detail extracellular basophilic deposits identifiable by Alcian blue staining at pH 2.7, often surrounded by macrophages, eosinophils, or neutrophils in a foreign body–type pattern [19]. However, our patient’s sample showed only subtle inflammatory changes rather than a robust granulomatous or fibrotic reaction, reinforcing a milder process consistent with HA filler-related PIH. Taken together, this clinicopathologic correlation indicates that subclinical inflammation from the HA filler, rather than a severe adverse response, is most likely responsible for the localized lip hyperpigmentation. In our patient, the fact that the hyperpigmentation was limited to the lower lip can be explained by the possibility of receiving higher doses of HA fillers in this area. The patient’s principal concern was malignancy; once histopathology excluded this, she declined active intervention and elected clinical observation with strict photoprotection.

The prevention of hyperpigmentation secondary to HA injections relies mostly on technical adjustments. PIH is avoided by limiting trauma (fine cannulas, deep injection), avoiding inflammatory fillers (LMW-HA), and imposing strict photoprotection. In cases of hyperpigmentation, which can present a complex therapeutic challenge, treatment options include topical depigmenting agents (hydroquinone, retinoids), chemical peels (alpha-hydroxy acids, trichloroacetic acid), and pigment lasers (Q-switched or picosecond). It is important to note that these treatments are not foolproof and should be tested on a small area first, as both chemical peels and laser procedures can themselves induce hyperpigmentation. Biofilm-related hyperpigmentation is prevented by rigorous asepsis and the use of sterile single-use equipment. If it occurs, prolonged antibiotic therapy can be indicated, with corticosteroids if needed. Neovascularization is prevented by avoiding traumatic injections and highly hydrophilic fillers in highly vascularized areas. The treatment relies on vascular lasers, sclerotherapy for persistent cases, and corticosteroids in case of inflammation. An approach tailored to each mechanism, a controlled injection, and post-procedure follow-up help limit these complications and optimize the tolerance of fillers [8,9]. A concise comparison of the clinical presentation, histopathology, timing, and management of drug-induced hyperpigmentation, filler-related post-inflammatory hyperpigmentation, and solar lentigo is summarized in Table 1.

## 5. Conclusions

Although filler-induced hyperpigmentation is a known side effect, only a few cases have been reported in the literature. Since the clinical appearance is not specific, and may be confused with other hyperpigmentation conditions, such as solar lentigo, or drug-induced hyperpigmentation, we think that the knowledge of the possibility to have hyperpigmented lesions following filler injections would be highly important for clinical dermatologists and dermatopathologists.

## Figures and Tables

**Figure 1 dermatopathology-12-00041-f001:**
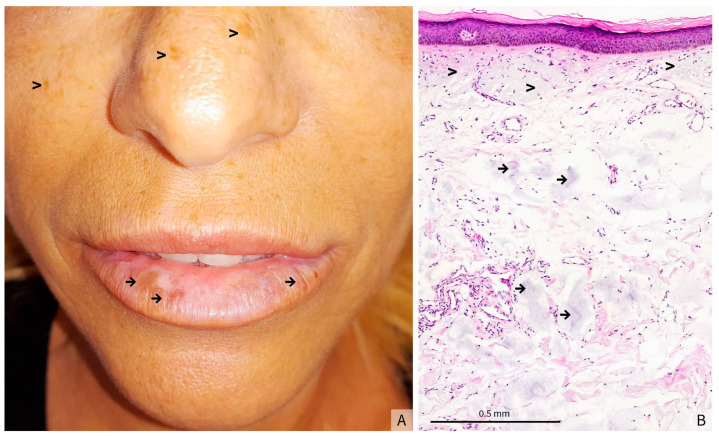
(**A**) Clinical presentation: hyperpigmented macules on the lower lip (→) and signs of chronic UV exposure (>), (**B**) (Hematoxylin-eosin stain, original magnification × 5) Atrophic and flattened epidermis. The underlying dermis shows prominent actinic elastosis (>), associated with amorphous grayish deposits interspersed throughout the mid and deep dermis (→), findings that may correspond to residual filler material.

**Figure 2 dermatopathology-12-00041-f002:**
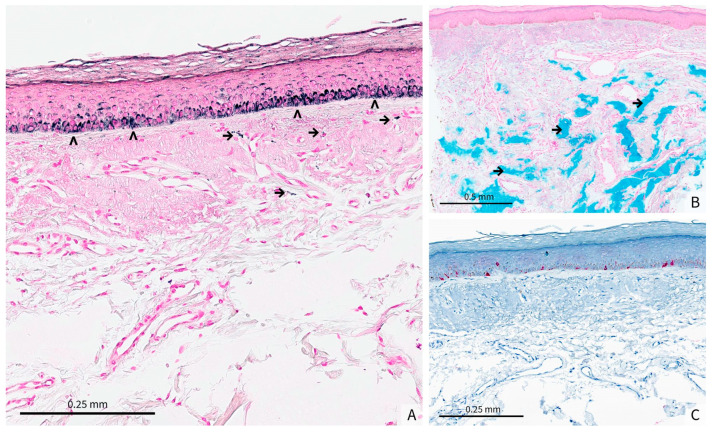
(**A**) (original magnification × 15) Fontana-Masson staining demonstrates a heterogeneous epidermal hyperpigmentation within the basal layer (>). Mild pigmentary incontinence is evident in the superficial dermis, with scattered melanin-laden macrophages (→), (**B**) (Original magnification × 5) Colloidal iron staining reveals blue, amorphous deposits throughout the dermis (→). These deposits are consistent with the presence of HA filler material, showing a patchy distribution that corresponds to the areas of clinical hyperpigmentation, (**C**) (Original magnification × 10) Melan-A immunostaining does not show melanocytic hyperplasia or atypical melanocytic proliferation.

**Table 1 dermatopathology-12-00041-t001:** Comparison of drug-induced hyperpigmentation, filler-related post-inflammatory hyperpigmentation (PIH), and solar lentigo—concise summary of typical distribution, timing, clinical appearance, key histologic features, and prevention/management points as presented in this report.

Feature	Drug-Induced Hyperpigmentation	Filler-Related PIH	Solar Lentigo
Distribution	Often patchy; can be widespread or mucosal	Focal, at injection site; irregular, asymmetric	Localized to chronically sun-exposed sites
Onset	Weeks–months after exposure	Weeks–months after injection (may be delayed)	Gradual over months–years
Colour/appearance	Variable (gray-blue to brown)	Light brown to grayish, localized	Light to dark brown macules
Key histology	Variable: dermal deposition, melanosis or pigment incontinence	Epidermal melanosis ± mild pigment incontinence; possible extracellular HA deposits	Epidermal hyperplasia with clubbing, Basal hyperpigmentation, solar elastosis
Prevention/treatment	—	Technical prevention; topical agents, peels, lasers; hyaluronidase if indicated	Photoprotection; topical lightening agents; cryotherapy or superficial chemical peels; pigment-targeted lasers

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
