# Peer review of "Asymmetric Lip Hyperpigmentation in a Transplant Patient"

_dermatopathology, 2025, doi:10.3390/dermatopathology12040041_

Round 1

Reviewer 1 Report

Comments and Suggestions for Authors

The authors describe a 56-yo female patient with asymmetric pigmentation of her lower lip. The colorwas yellowish brown thus excliding most of the later discussed differential diagnoses.

Fig 1 is twice, the second should be designated fig 2.

The authors show colloidal iron stain whicgh is very intense over the HA filler. A word of explanation should be given.

"... degradation of HA, particularly low molecular weight fragments (LMW-HA), has also been associated with inflammation through the activation of macrophages and dendritic cells, leading to a release of pro-inflammatory cytokines that may promote pigmentation" The histopathology does not show any inflammation so I think this is not plausible explanation.

". The Tyndall effect is avoided by a deep injection and an appropriate choice of filler type" The Tyndall effect never gives the light brown color seen inthis patient.

" Its treatment relies on hyaluronidase to dissolve the product and, if necessary, on pigment lasers (Q-switched or picosecond)". I agree with hyaluronidase, but completely disagree with lasers for the Tyndall effect. And why the picosecond laser?

Author Response

  1. The authors describe a 56-yo female patient with asymmetric pigmentation of her lower lip. The colorwas yellowish brown thus excliding most of the later discussed differential diagnoses.
    -- We thank the reviewer for this observation. The lesion’s yellowish-brown hue was noted on clinical inspection. However, clinical colour alone is not reliably specific and can overlap across several entities (drug-related pigmentation, actinic change, and filler-related post-inflammatory change). For that reason we retained the listed differentials pending histological evaluation; the histopathology (epidermal melanosis with mild pigment incontinence and absence of granulomatous or fibrotic change) ultimately supported a focal post-filler inflammatory mechanism. We therefore believe the current diagnostic reasoning in the text is appropriate and do not propose additional changes, but are happy to add clarifying wording if the editors require it.

  2. Fig 1 is twice, the second should be designated fig 2.
    -- We thank the reviewer for this observation. This has been corrected.

  3. The authors show colloidal iron stain which is very intense over the HA filler. A word of explanation should be given.
    -- We thank the reviewer for this observation. An explanatory sentence has been added to the text (lines 53-56).

  4. "... degradation of HA, particularly low molecular weight fragments (LMW-HA), has also been associated with inflammation through the activation of macrophages and dendritic cells, leading to a release of pro-inflammatory cytokines that may promote pigmentation" The histopathology does not show any inflammation so I think this is not plausible explanation.
    -- We thank the reviewer for this comment. Pigment incontinence indicates a prior epidermal→dermal pigment leakage and can reflect resolved inflammation even when a cellular infiltrate is absent on biopsy. LMW-HA remains a plausible trigger for transient inflammation but is not presented as proven in this case; we have softened the wording. We added the clarifying sentence: “It is important to note that post-inflammatory hyperpigmentation can still be present clinically and histologically after the inflammatory infiltrate has subsided.” (128-130)

  5. The Tyndall effect is avoided by a deep injection and an appropriate choice of filler type" The Tyndall effect never gives the light brown color seen in this patient.
    -- We thank the reviewer for this observation. We mentioned the Tyndall effect only to enumerate possible mechanisms of filler-related discoloration; it was not considered a likely cause in this case. Histology and the clinical colour are inconsistent with a Tyndall phenomenon. We added a sentence in that sense in the manuscript: "The Tyndall effect and neovascularization were not considered compatible with the clinical presentation in our patient." (lines 143-144)

  6. " Its treatment relies on hyaluronidase to dissolve the product and, if necessary, on pigment lasers (Q-switched or picosecond)". I agree with hyaluronidase, but completely disagree with lasers for the Tyndall effect. And why the picosecond laser?
    -- We thank the reviewer for this comment. We agree that the Tyndall effect should be managed primarily by hyaluronidase to dissolve superficially placed product, and that laser therapy is not an appropriate treatment for Tyndall discoloration. We have removed the previous wording implying laser treatment for the Tyndall effect and clarified the text accordingly. (189-190)

Reviewer 2 Report

Comments and Suggestions for Authors

The authors present a case report where a female patient was administered to clinic with a hyperpigmentation in her lower lip. The hyperpigmentation appeared as irregular dark spots without bleeding or discomfort.

The patient has received a kidney and liver transplant, while she had also received two injections of hyaluronic acid in her lip. The patient received long-term immunosuppressive therapy.

Histological analysis was carried out to assess the condition. Histology showed slightly atrophic epidermis and dermal actinic elastosis. Colloidal iron staining revealed HA deposits. Further, absence of melanocytic hyperplasia or atypic melanocytic proliferation was demonstrated.

The diagnosis was post-filler hyperpigmentation of the lip. The authors describe in detail their conclusion based on the histological findings.

The case report is interesting, and I recommend its publication after responding to the following minor points.

Minor points

  1. There is a problem with Figure numbering. All figures are labelled as Figure 1, and on page 2 reference is given to Figure 2A (line 49) and Figure 2B (line 50) and Figure 3A (line 52) and Figure 3B (line 53). Please fix it.
  2. Please add bars with dimensions in biopsies.
  3. The authors need to report whether the injection of HA was performed only in the lower lip.

Author Response

  1. There is a problem with Figure numbering. All figures are labelled as Figure 1, and on page 2 reference is given to Figure 2A (line 49) and Figure 2B (line 50) and Figure 3A (line 52) and Figure 3B (line 53). Please fix it.
    -- We thank the reviewer for this observation. This has been corrected.

  2. Please add bars with dimensions in biopsies. 
    -- We thank the reviewer for this suggestion. We have added dimension bars in the figures. 

  3. The authors need to report whether the injection of HA was performed only in the lower lip. 
    -- We thank the reviewer for this request for clarification. As stated in the initial manuscript, hyaluronic acid injections were performed in both lips (line 29). We can make this more explicit if the editors wish.

Reviewer 3 Report

Comments and Suggestions for Authors

This manuscript describes a case of lip hyperpigmentation following hyaluronic acid (HA) filler injection. The authors confirmed the presence of HA with colloidal iron staining and discussed localized inflammation as a possible cause of the pigmentation. This is a unique case, and reporting it has educational value in raising awareness among readers. However, before acceptance, the following issues should be clarified.

Major comments

  1. Lack of clinical course and therapeutic information
    • It should be clarified why filler injection was performed in a patient under immunosuppressive therapy after organ transplantation.
    • Please indicate whether filler injections were also performed in the upper lip.
    • The authors should briefly mention whether any treatment was attempted for the hyperpigmentation, as this information would be useful.
  2. Association with organ transplantation
    • The title emphasizes the special background of an “organ transplant patient,” but the Discussion scarcely addresses the possible relationship between organ transplantation, immunosuppression, and pigmentation. The authors should at least briefly discuss whether immunosuppression could influence inflammation or pigmentation in this context.
  3. Interpretation of clinical findings
    • The clinical image appears to show hypopigmentation in the central lower lip. Could filler injections cause not only hyperpigmentation but also hypopigmentation? The authors should provide their opinion.

Minor comments

  • In the Abstract, the abbreviation “HA” should be defined at its first appearance (as “hyaluronic acid”).

Author Response

  1. It should be clarified why filler injection was performed in a patient under immunosuppressive therapy after organ transplantation.
    -- We thank the reviewer for this comment. We have added a sentence in the manuscript : "In the corresponding period, she was taking immunosuppressive drugs, including prednisone, everolimus, tacrolimus and ciclosporin." (lines 30-31)

  2. Please indicate whether filler injections were also performed in the upper lip.
    -- We thank the reviewer for this request for clarification. As stated in the initial manuscript, hyaluronic acid injections were performed in both lips (line 29). We can make this more explicit if the editors wish.

  3. The authors should briefly mention whether any treatment was attempted for the hyperpigmentation, as this information would be useful. 
    -- We thank the reviewer for this comment. We have added a sentence in the manuscript : "The patient’s principal concern was malignancy; once histopathology excluded this, she declined active intervention and elected clinical observation with strict photoprotection." (lines 175-177)

  4. The title emphasizes the special background of an “organ transplant patient,” but the Discussion scarcely addresses the possible relationship between organ transplantation, immunosuppression, and pigmentation. The authors should at least briefly discuss whether immunosuppression could influence inflammation or pigmentation in this context.
    -- We thank the reviewer for this important remark. We have addressed it with a brief paragraph in the Discussion (lines 145-157) that considers how chronic immunosuppression may have modulated the local response to filler injection. 

  5. The clinical image appears to show hypopigmentation in the central lower lip. Could filler injections cause not only hyperpigmentation but also hypopigmentation? The authors should provide their opinion. 
    -- We thank the reviewer for this observation. We have addressed it in the manuscript (lines 160–164): “The lighter appearance of the central vermilion most likely reflects the ‘normal’ lip color associated with major actinic damage and solar elastosis as seen in Figure 1B, rather than a true hypopigmentation, as we did not observe a decrease in epidermal melanin with the Fontana-Masson stain at the periphery of the biopsy.” In addition, histology showed no epidermal necrosis, scarring, or other features (for example vascular occlusion) that would suggest true post-procedural loss of melanocytes. On this basis we interpret the central lighter zone as relative (pseudo) hypopigmentation due to contrast with adjacent hyperpigmented skin rather than a separate hypopigmentary complication. 

  6. In the Abstract, the abbreviation “HA” should be defined at its first appearance (as “hyaluronic acid”). 
    -- We thank the reviewer for this comment. We have corrected this.

Reviewer 4 Report

Comments and Suggestions for Authors

This is an interesting and well documented case. The manuscript is concise, well structured, and supported by clear histopathological correlation. It provides valuable insights for clinicians and pathologists.

The discussion would benefit from a summary table comparing drug-induced hyperpigmentation vs. filler-related PIH vs. solar lentigo to emphasize their features.

If possible, please include details on clinical course and management.

The references are relevant, yet older sources (the 1987 study) could be balanced with more recent systematic reviews.

Author Response

  1. The discussion would benefit from a summary table comparing drug-induced hyperpigmentation vs. filler-related PIH vs. solar lentigo to emphasize their features. 
    -- We thank the reviewer for this constructive suggestion. A concise comparative table has now been added to the Discussion (Table 2), summarizing distribution, timing, clinical appearance, key histologic features and prevention/treatment for drug-induced hyperpigmentation, filler-related post-inflammatory hyperpigmentation and solar lentigo. The table is derived solely from material already presented in the manuscript.

  2. If possible, please include details on clinical course and management. 
    -- We thank the reviewer for this suggestion. We have added a brief statement to the Clinical Course section (lines 175-177) clarifying that the patient’s principal concern was malignancy; after histopathology excluded this, she declined active intervention and elected clinical observation with strict photoprotection. No topical, enzymatic, laser or surgical treatments were performed.

  3. The references are relevant, yet older sources (the 1987 study) could be balanced with more recent systematic reviews. 
    -- Thank you for this comment. We have replaced the older thiazide reference (1987) with a more recent review (Blakely et al., 2019) that specifically addresses hydrochlorothiazide-associated photosensitivity and summarizes current clinical patterns and mechanisms. This substitution provides updated support for the statements in the paragraph while leaving the text otherwise unchanged. If the editors would like additional recent sources cited here, we are happy to add them.